# The Role of LincRNA-EPS/Sirt1/Autophagy Pathway in the Neuroprotection Process by Hydrogen against OGD/R-Induced Hippocampal HT22 Cells Injury

**DOI:** 10.3390/jpm13040631

**Published:** 2023-04-03

**Authors:** Ya-Hong Li, Shun Zhang, Lu Tang, Jianguo Feng, Jing Jia, Ye Chen, Li Liu, Jun Zhou

**Affiliations:** 1Department of Anesthesiology, The Affiliated Hospital of Southwest Medical University, Luzhou 646000, China; 2Anesthesiology and Critical Care Medicine Key Laboratory of Luzhou, The Affiliated Hospital of Southwest Medical University, Luzhou 646000, China; 3Department of Traditional Chinese Medicine, The Affiliated Hospital of Southwest Medical University, Luzhou 646600, China

**Keywords:** OGD/R, HT22 cells, hydrogen, LincRNA-EPS, Sirt1, autophagy

## Abstract

Cerebral ischemia/reperfusion (CI/R) injury causes high disability and mortality. Hydrogen (H_2_) enhances tolerance to an announced ischemic event; however, the therapeutic targets for the effective treatment of CI/R injury remain uncertain. Long non-coding RNA lincRNA-erythroid prosurvival (EPS) (lincRNA-EPS) regulate various biological processes, but their involvement in the effects of H_2_ and their associated underlying mechanisms still needs clarification. Herein, we examine the function of the lincRNA-EPS/Sirt1/autophagy pathway in the neuroprotection of H_2_ against CI/R injury. HT22 cells and an oxygen-glucose deprivation/reoxygenation (OGD/R) model were used to mimic CI/R injury in vitro. H_2_, 3-MA (an autophagy inhibitor), and RAPA (an autophagy agonist) were then administered, respectively. Autophagy, neuro-proinflammation, and apoptosis were evaluated by Western blot, enzyme-linked immunosorbent assay, immunofluorescence staining, real-time PCR, and flow cytometry. The results demonstrated that H_2_ attenuated HT22 cell injury, which would be confirmed by the improved cell survival rate and decreased levels of lactate dehydrogenase. Furthermore, H_2_ remarkably improved cell injury after OGD/R insult via decreasing pro-inflammatory factors, as well as suppressing apoptosis. Intriguingly, the protection of H_2_ against neuronal OGD/R injury was abolished by rapamycin. Importantly, the ability of H_2_ to promote lincRNA-EPS and Sirt1 expression and inhibit autophagy were abrogated by the siRNA-lincRNA-EPS. Taken together, the findings proved that neuronal cell injury caused by OGD/R is efficiently prevented by H_2_ via modulating lincRNA-EPS/Sirt1/autophagy-dependent pathway. It was hinted that lincRNA-EPS might be a potential target for the H_2_ treatment of CI/R injury.

## 1. Introduction

Cerebral ischemic injury is a major contributor to death or different degrees of disability worldwide. Cerebral ischemic injury may result from several different events, including external forces and degradation of the blood vessels, as well as thrombotic or thromboembolic arterial occlusions, etc. [1,2,3,4]. Clinically, cardiopulmonary resuscitation from cardiac arrest is among the leading contributors to cerebral ischemia-reperfusion (CI/R) injury. The timely restoration of blood flow is the key to treating cerebral ischemia. However, reperfusion after ischemia has been accompanied by a series of pathological reactions, which could cause permanent damage to brain tissue, and these reactions include the production of oxygen radicals, an excessive amount of calcium ions within the cell, glutamate-induced neurotoxicity, inflammatory response, cell apoptosis, and autophagy [2,5,6]. Therefore, the intervention of the above-mentioned pathological reactions is of great importance to improve CI/R injury. A series of protective strategies have been studied to alleviate CI/R injury; however, the benefits have not yet been clinically applicable, owing to late delivery, low potency, or poor tolerability. Interestingly, postconditioning (PoCo) is clinically more applicable under conditions in which there is an unannounced CI/R event.

Hydrogen is a small molecule that diffuses across cell membranes rapidly and has shown no toxicity in humans [7]. Numerous research reports have illustrated that hydrogen possesses anti-inflammatory, antioxidant, anti-allergic, and anti-apoptotic properties, and it can regulate autophagy levels [7,8]. The protective effects of hydrogen have been demonstrated on a variety of diseases [9,10,11,12,13]. To date, our previous research demonstrated that the injection of hydrogen can reduce the brain’s oxidative stress and neuronal apoptosis, improve neurological function, and enhance survival rates of rabbit models of cardiac arrest [14]. Other studies showed that hydrogen can alleviate CI/R injury in the hippocampal region and improve neurological deficit scores and survival rates in rat models of cardiac arrest [15,16]. Although multiple research reports have proven the protective effects of hydrogen conditioning (H_2_) against CI/R injury, the underlying mechanisms have remained elusive.

Long non-coding RNA (lncRNA) is a class of RNAs with molecules longer than 200 nucleotides that lack an appreciable open reading frame, accounting for the majority of noncoding RNAs [17]. Studies showed that lncRNAs regulate the expression levels of protein-coding genes at various levels, such as epigenetic, transcription, and post-transcriptional levels, and are involved in the regulation of multiple activities, including the silencing of the X chromosome, genome imprinting, chromatin modification, transcriptional activation, transcriptional interference, and intranuclear transport [17,18,19,20]. Atianand et al. found that long intergenic noncoding RNA erythroid prosurvival (lincRNA-EPS) functions as a transcription brake to suppress inflammatory responses [21]. Subsequently, it was recently reported that lincRNA-EPS can inhibit microglial activation, cytotoxicity, and apoptosis, and elevates the proliferative rate of neural stem cells in vitro. In animal models of transient middle cerebral artery occlusion, lincRNA-EPS reduced the permeability and cytotoxicity of inflammatory cells and improved neuronal death in ischemic regions [22]. Another study found that lincRNA-EPS inhibited autophagy in Bacillus Calmette-Guerin (BCG)-infected RAW264.7 macrophage cells by downregulating the JNK/MAPK signaling pathway [23]. Based on these data, lincRNA-EPS may be involved in the regulation of neurological function by modulating multiple signaling pathways, but the specific mechanism has not yet been clarified.

Silent information regulator 1 (Sirt1) is a deacetylase that depends on nicotinamide adenine dinucleotide (NAD^+^), which is implicated in the regulation of a broad variety of biological processes such as autophagy in I/R injury, cell apoptosis, mitochondrial biogenesis, inflammatory response, and energetic homeostasis [24,25]. A previous research report illustrated that hydrogen reduces hepatocyte apoptosis by activating the HO-1/Sirt1 pathway against hepatic I/R injury [26]. Mei et al. found that electroacupuncture ameliorates CI/R injury by inhibiting the autophagic process via the Sirt1-FoxO1 signaling pathway [27]. Autophagy is the mechanism by which cytoplasmic components are degraded in a manner that is dependent on lysosomes [28]. Autophagy includes three types: macro-autophagy, micro-autophagy, and chaperone-mediated autophagy, occurring through the formation of autophagosomes along with a subsequent association with lysosomes, leading to the formation of the autophagolysosome complex. The low autophagy level was physiologically confirmed, while in a stress state, autophagy was enhanced, and it exerted degrading effects [29,30]. A protective function of moderate autophagy in CI/R injury by degrading damaged organelles and misfolded proteins was previously reported [31]. However, excessive autophagy might lead to excessive normal protein and cellular organelle degradation, eventually exacerbating cell injury and even causing cell death [32]. Therefore, treatment strategies that target autophagy as a mechanism for treating CI/R injuries show great promise.

To further understand how H_2_ protects neurons from CI/R damage, we conducted an in vitro investigation to examine the function of the lincRNA-EPS/Sirt1/autophagy pathway.

## 2. Materials and Methods

### 2.1. Cell Culture

HT22 cell, an immortalized mouse hippocampal neuronal cell line, was subcloned from parent HT4 cells originally immortalized from cultures of primary mouse hippocampal neurons [33]. HT22 cell was obtained from the Laboratory of Anesthesiology, Southwest Medical University (Luzhou, China). HT22 cells were cultured in a humid incubator at 37 °C containing 5% CO_2_ and 95% air environment in Dulbecco’s modified Eagle’s medium (DMEM) containing 10% fetal bovine serum (FBS) and 100 U penicillin/streptomycin.

### 2.2. OGD/R Injury Model Development

The induction of OGD/R injury in HT22 cells was performed as reported in previous studies [32,34,35]. After being rinsed thrice in phosphate-buffered saline (PBS), the cells were cultured in a serum-free, glucose-free DMEM culture. Immediately after harvesting, the cells were introduced in a tri-gas incubator (Eppendorf, Hamburg, Germany) for 24 h at 37 °C for 2-, 4-, 6-, 8, or 12 h with 4% N_2_/5% CO_2_/1% O_2_. Following these procedures, the cells were returned to an oxygenated, glucose-containing DMEM under normoxic conditions for 24 h (reoxygenation). In addition, 3-methyladenine (3-MA) or rapamycin (RAPA) was added into the medium at the beginning of reoxygenation and left until the end of reoxygenation.

### 2.3. Synthesis of a Medium Enriched with Hydrogen

The procedure for creating a hydrogen-rich medium has already been detailed [36]. In brief, hydrogen was dissolved in a DMEM at a pressure of 0.4 MPa. Then, the concentration of hydrogen in the solution was determined using a hydrogen electrode, and when the H_2_ concentration reached 0.6 mmol/L, the preparation of a medium enriched with hydrogen was completed. The medium was freshly produced to guarantee high H_2_ concentration. In the H_2_ group, a hydrogen-rich medium was used to culture the cells, rather than the normal medium at the beginning of the reoxygenation.

### 2.4. Experimental Design

The HT22 cells were classified for the following tests in the current investigation.

To find out an optimal OGD time, these six groups of cells were created by a random sampling process: (1) control (Con) group—except for not performing OGD/R operation, all other operational steps were the same as those in the OGD/R group; (2) OGD-2h/R group—OGD for 2 h plus 24 h of reoxygenation; (3) OGD-4h/R group—OGD for 4 h plus 24 h of reoxygenation; (4) OGD-6h/R group—OGD for 6hrs plus 24 h of reoxygenation; (5) OGD-8h/R group—OGD for 8 h plus 24 h of reoxygenation; (6) OGD-12h/R group—OGD for 12 h plus 24 h of reoxygenation.

The cells were randomized into the following 4 groups to examine the protective properties of H_2_ on OGD/R injury in HT22 cells: (1) control (Con) group; (2) hydrogen (Con+ H_2_) group—except for not performing OGD/R operation, all other operational steps were as same as those in the OGD/R+ H_2_ group; (3) OGD/R group; (4) OGD/R+ H_2_—cells were cultured in a medium enriched with H_2_ rather than the normal medium at the beginning of the reoxygenation, and all other operational steps were the same as those in the OGD/R group.

To additionally examine the underlying protective mechanism of H_2_, we utilized 3-MA and RAPA to inhibit autophagy in cells and used small interfering RNAs (siRNAs) to knock down intracellular lincRNA-EPS. First, to examine the involvement of autophagy in the protective function of H_2_ on OGD/R injury in HT22 cells, the cells were randomized into 6 groups as indicated: (1) OGD/R group; (2) OGD/R+ H_2_ group; (3) OGD/R+3-MA group—10 mM 3-MA (J&K Scientific, Beijing, China) was introduced into the normal medium at the beginning of the reoxygenation; (4) OGD/R+ H_2_+3-MA group—10 mM 3-MA was added into the hydrogen-rich medium at the beginning of the reoxygenation; (5) OGD/R+RAPA group—500 nM RAPA (Absin, Shanghai, China) was added into the normal medium at the beginning of the reoxygenation; (6) OGD/R+ H_2_+RAPA group—500 nM RAPA was added into the H_2_-rich medium at the beginning of the reoxygenation. Then, to confirm whether H_2_ could exert a cytoprotective effect by activating intracellular lincRNA-EPS to inhibit autophagy following OGD/R injury, the cells were randomized into the following six groups: (1) OGD/R group; (2) OGD/R+ H_2_ group; (3) OGD/R+siRNA-NC group—siRNA-NC was transfected into the cells 48hrs before OGD/R.; (4) OGD/R+siRNA group—siRNA-lincRNA-EPS was used to transfect the cells 48hrs before OGD/R; (5) OGD/R+ H_2_+siRNA-NC group; (6) OGD/R+ H_2_+siRNA group.

### 2.5. Cell Viability and Cytotoxicity Assays

The 3-(4,5-dimethylthiazol-2-yl)-2,5-diphenyl tetrazolium bromide (MTT) test was conducted to determine whether or not HT22 cells were viable [37]. In total, 200 μL of growth medium was used to seed cells into 96-well plates (5 × 10^3^ cells/well), after which they were incubated overnight and treated according to different experimental groups. A 24 h reoxygenation period was followed by the addition of 10 μL of MTT solution per well and a 4-h incubation period at 37 °C. A volume of 100 μL of dimethyl sulfoxide (DMSO) was added to the culture after the media had been gently aspirated. Afterward, spectrophotometer readings were taken at 570 nm to measure each well’s optical density (OD).

Using a commercially available kit (Jiancheng Bioengineering Institute, Nanjing, China), a lactate dehydrogenase (LDH) test was conducted to identify the HT22 cells’ membrane permeability following the directions stipulated by the manufacturer.

### 2.6. Cell Apoptosis Assay

For the analysis of cell apoptosis, an Annexin V-FITC/PI kit (BD Biosciences, Franklin Lakes, NJ, USA) was utilized. In brief, following the seeding of 6-well plates with HT22 cells (1 × 10^5^ cells/well), they were cultured for a full night, and treated according to different experimental groups. After collecting the cells using trypsin without EDTA, they were rinsed thrice using PBS. Then, following the resuspension of the cells in binding buffer, they were stained with 5 μL of Annexin-V FITC and propidium iodide (PI), and then they were subjected to incubation for 15 min in darkness at room temperature (RT), as instructed by the manufacturer. Finally, the apoptotic rate was determined by a flow cytometer (BD Biosciences) [38].

### 2.7. Detection of Inflammatory Factors

Cell culture supernatant was collected after OGD/R. Then, The enzyme-linked immunosorbent assay (ELISA) kits (Meilian Co., Ltd., Wuhan, China) were used to measure the levels of tumor necrosis factor-α (TNF-α) and interleukin-1β (IL-1β) that were present in the supernatant of cell cultures in compliance with the directions provided by the manufacturer as described previously [39].

### 2.8. Tandem mCherry-EGFP-LC3 Immunofluorescence

Lenti-mCherry-EGFP-LC3B (Beyotime Institute of Biotechnology, Shanghai, China) was introduced into HT22 cells via transfection [40]. After puromycin selection for positive transfection, EGFP-positive HT22 cells were visualized by a microscope, and EGFP-positive colonies were harvested and expanded for subsequent experiments. Before treatment, HT22-mCherry-EGFP-LC3B cells were grown throughout the night on glass coverslips in a DMEM containing 10% FBS. After the treatment, a 4% paraformaldehyde (PFA) solution was applied to fix the cells on the coverslips for half an hour in darkness at RT before rinsing them thrice in PBS, labeling them with 4′,6-diamidino-2-phenylindole (DAPI) for 6–8 min, and rinsing again using PBS. An upright fluorescent microscope (Nikon, Tokyo, Japan) was used to observe the cells’ EGFP and mCherry fluorescence. The nucleus was visualized with the help of DAPI staining. The ratio of the fluorescence of mCherry+ puncta to the fluorescence of EGFP+ puncta was used as the autophagic index. Cells were photographed and scored in a blind manner utilizing the ImageJ program (National Institutes of Health, Bethesda, MD, USA). More than 50 cells were counted for each condition.

### 2.9. siRNA Transfection

The siRNA-RNA-EPS was transfected with the riboFECTTM transfection kit (166T; Ribobio Co., Ltd., Guangzhou, China) into the HT22 cells as per the package recommendations. In addition, siRNA control was used as a negative control (siRNA-NC). The cells were employed for further study or experimentation 48 h following transfection. Below is the list of sequences that were applied: siRNA1-lincRNA-EPS:forward: 5′- GGUUUAGCACUCACUGCUAGC -3′, Reverse: 5′- UAGCAGUGAGUGCUAAACCGU -3′; siRNA2-lincRNA-EPS: forward: 5′- CGCAUGGUCACUCACCUAAUA -3′, Reverse: 5′- UUAGGUGAGUGACCAUGCGUG -3′; siRNA3-lincRNA-EPS: forward: 5′- CAUGGUCACUCACCUAAUAAG -3′, Reverse: 5′- UAUUAGGUGAGUGACCAUGCG -3′, and siRNA-NC (lnc6N0000001-1-10; Ribobio Co., Ltd.) [23].

### 2.10. Real-Time Polymerase Chain Reaction (RT-PCR)

Total RNA was isolated from the HT22 cells using the TRIzol reagent (Tiangen, Beijing, China), as per the package directions. Afterward, we utilized a HiScrip III All-in-one RT SuperMix Perfect for qPCR kit (Vazyme Biotech Co., Ltd., Nanjing, China) to reverse transcribe 1 μg of the extracted total RNA. RT-PCR was conducted utilizing a Taq Pro Universal SYBR kit (Vazyme Biotech Co., Ltd.) in a real-time PCR system (Roche, Basel, Switzerland). An initial incubation at 95 °C for 15 min was accompanied by 5 cycles of incubation at 95 °C for 10 s and at 60 °C for 32 s to accomplish the amplification. The procedure started with a 30 s denaturation at 95 °C, followed by 40 cycles of 10 s denaturation at 95 °C and 30 s annealing at 60 °C. Melting curve analysis was conducted at 95 °C for 15 s, and annealing at 60 °C for 60 s and at 95 °C for 15 s as directed by the manufacturer. Table 1 displays the RT-PCR primer sequences. Glyceraldehyde 3-phosphate dehydrogenase (GAPDH) was chosen to serve as the internal reference. Additionally, the 2^−ΔΔCT^ method was utilized to measure the relative mRNA levels. Target gene mRNA expression was represented as a fold-change compared with the Con or OGD/R group [39].

### 2.11. Western Blot Assay

Following the reoxygenation process, the HT22 cells were collected for analysis. RIPA lysis solution (Beyotime Institute of Biotechnology) containing 1% phenylmethylsulfonyl fluoride (PMSF) was utilized to lyse these cells for half an hour on ice. Cell lysates were centrifuged at a rate of 14,000× *g* for 10 min at 4 °C, and the protein concentration of the supernatant was quantified using the bicinchoninic acid assay (BCA) kit (Beyotime Institute of Biotechnology). In addition, 20 µg of protein was loaded, electrophoresed on the sodium dodecyl sulfate-polyacrylamide gel electrophoresis (SDS-PAGE) gel, transferred to a polyvinylidene fluoride (PVDF) membrane (Amersham Biosciences, NJ, USA) or a nitrocellulose membrane (Beyotime Institute of Biotechnology), blocked with 5% skim milk for 1 h, and probed at 4 °C overnight with the following primary antibodies: anti-Sirt1 (dilution, 1:1000; Cell Signaling Technology (CST), Danvers, MA, USA), anti-Beclin1 (dilution, 1:1000; CST), anti-LC3-II (dilution, 1:1000; CST), and anti-β-actin (dilution, 1:3000; Proteintech, Rosemont, IL, USA). The next step involved incubating the sample for 1 h at 37 ° C with a secondary antibody conjugated to horseradish peroxidase (dilution, 1:5000; Absin, Shanghai, China). Eventually, the membrane was subjected to enhanced chemiluminescence (ECL), after which the immunoreactive bands were evaluated with the use of the ImageJ 1.31 program (National Institutes of Health). The data were normalized using beta-actin as the internal control [39].

### 2.12. Statistical Analysis

The analyses of all statistical data were conducted with the use of GraphPad Prism 8.3 statistical software (GraphPad Software Inc., San Diego, CA, USA). Means ± standard deviations (SD) were used to present all data. One-way analysis of variance (ANOVA) and Tukey’s post-hoc test were utilized for making comparisons. A *p* < 0.05 denoted the significance criterion.

## 3. Results

### 3.1. The Damage of HT22 Cells Was Aggravated after OGD/R

The survival rate of the HT22 cells gradually decreased with an increase in the duration of OGD treatment, while the amount of LDH produced increased. It was discovered that each OGD/R group had significantly worse HT22 cell injury than the Con group. Based on past research and the fact that only approximately half as many HT22 cells survived in the OGD-6h/R group as did in the Con group, we decided to use the OGD-6h/R group for our subsequent experiments to allow for moderate damage to HT22 cells (Figure 1).

### 3.2. H_2_ Protected HT22 Cells from OGD/R Injury

To determine whether H_2_ could protect HT22 cells against damage elicited by OGD/R, the cells were exposed to the hydrogen-rich medium at the beginning of the reoxygenation. When compared with the Con group, the cell survival rate in the OGD/R group was reduced, while the LDH level, the levels of pro-inflammatory factors (TNF-α and IL-1β), and the cellular apoptotic rate were elevated. Additionally, the cell survival rate in the OGD/R+ H_2_ group increased, while the LDH level, the levels of pro-inflammatory factors (TNF-α and IL-1β), and the cell apoptosis rate decreased relative to those of the Con group (Figure 2). Notably, the variation between the Con group and the Con+ H_2_ group was insignificant.

### 3.3. H_2_ Upregulated lincRNA-EPS and Sirt1, and Inhibited Autophagy

The possible protective mechanism of H_2_ against neuronal OGD/R injury was explored. We discovered that the lincRNA-EPS as well as both Beclin-1 and LC3-II mRNA and protein levels increased remarkably, while the Sirt1 protein and mRNA levels were decreased in the OGD/R group in contrast to those in the Con group. Meanwhile, H_2_ upregulated the lincRNA-EPS and Sirt1 mRNA and protein levels while decreasing the Beclin-1 and LC3-II mRNA and protein levels in the OGD/R damage group relative to the Con group (Figure 3). These data indicated that H_2_ could improve neuronal OGD/R damage by suppressing autophagy and upregulating the lincRNA-EPS and Sirt1.

### 3.4. H_2_ Protected Neurons against OGD/R Injury by Suppressing Autophagy

We further explored the underlying protective mechanism of H_2_ using 3-MA and RAPA. First, we used some methods to detect neuronal damage. Figure 4 illustrates that the cell survival rate was higher in the OGD/R+ H_2_ group compared to that in the OGD/R+ H_2_+RAPA group. Meanwhile, H_2_ decreased the levels of LDH and pro-inflammatory factors in the medium, while attenuating the cell apoptosis rate, which could be abolished by RAPA during OGD/R.

Beclin1 and LC3-II protein and mRNA levels were then detected by Western blot analysis and RT-PCR. As illustrated in Figure 5, compared with the OGD/R group, the protein and mRNA levels of Beclin1 and LC3-II were lowered by H_2_ and were further reduced by 3-MA, while they were upregulated by RAPA during OGD/R.

### 3.5. The Changes of Autophagic Flux after OGD/R Injury

We began by transfecting HT22 cells with mCherry-EGFP-LC3B before triggering autophagy, then we examined the cell images to determine the autophagic flux alterations. According to the instructions, following the autophagosome-lysosome fusion, EGFP was secreted from mCherry-EGFP-LC3B and degraded in the lysosomes. Changes in fluorescence color from green-red (yellow) to red indicate the production of autolysosomes, a loss in the EGFP signal (green), and a reservation of the mCherry signal (red); these fluctuations are used to measure the autophagic flux. In our research, we used DAPI staining to visualize the nucleus. The results showed that H_2_ inhibited autophagy in the HT22 cells. Autophagy in the HT22 cells was remarkably enhanced after treatment with RAPA, and this effect was further amplified when RAPA was combined with H_2_ (Figure 6).

Collectively, based on these findings, H2 protects neurons from OGD/R injury by suppressing autophagy.

### 3.6. LincRNA-EPS Knockdown Abolished the Protection of H_2_ against Cell Injury

According to the previous results, it was revealed that H_2_ upregulated lincRNA-EPS and Sirt1 while inhibiting autophagy under OGD/R injury. However, whether H_2_ could inhibit autophagy and upregulate Sirt1 through activating lincRNA-EPS should be clarified. Thus, to examine the function of lincRNA-EPS in the protective ability of H_2_ against neuronal OGD/R-induced autophagy and injury, HT22 cells were transfected with siRNA-lincRNA-EPS.

Firstly, the RT-PCR was conducted to examine the knockdown effect of siRNA-lincRNA-EPS after transfection. Figure 7A illustrates that in contrast to the siRNA-NC group, the expression of lincRNA-EPS in the other three siRNA groups was significantly downregulated. Among them, the siRNA2-lincRNA-EPS had the most obvious knockdown effect; therefore, we selected siRNA2-lincRNA-EPS for follow-up experiments. We then detected some indicators to evaluate the neuronal injury. As displayed in Figure 7B–G, H_2_ inhibited neuronal OGD/R damage. The cell survival rate was higher, while the levels of LDH and pro-inflammatory factors in the medium and cell apoptosis rate were reduced in the OGD/R+ H_2_ group as opposed to those in the OGD/R group. Nevertheless, the protective effects of H_2_ were reversed by siRNA-lincRNA-EPS.

### 3.7. LincRNA-EPS Mediated the Effects of H_2_ on Sirt1 Expression and Autophagy

Western blot analysis and RT-PCR were subsequently conducted to determine the levels of Sirt1, Beclin1, and LC3-II expression. Figure 8 shows that H_2_ elevated Sirt1 mRNA and protein levels. However, siRNA-lincRNA-EPS combined with H_2_ substantially decreased the Sirt1 protein and mRNA levels. Moreover, Beclin1 and LC3-II levels in the OGD/R+ H2 group were elevated, as opposed to those in the OGD/R group, and these variations could be reversed by administering siRNA-lincRNA-EPS.

### 3.8. LincRNA-EPS Mediated the Effects of H_2_ on Autophagic Flux

Stable transfection of HT22 cells with tandem mCherry-EGFP-LC3B facilitated us to examine the impact of lincRNA-EPS knockdown on autophagic flux in response to OGD/R damage. As illustrated in Figure 9, H_2_ inhibited autophagy in HT22 cells, which was abolished by siRNA-lincRNA-EPS.

Collectively, these results indicated that H_2_ inhibits autophagy and activates Sirt1 by upregulating lincRNA-EPS under neuronal OGD/R injury.

## 4. Discussion

It is known that the weight of the human brain makes up only around 2–3% of the body’s total mass, but it consumes about 20% of the body’s oxygen supply. When an arterial occlusion occurs, it leads to ischemic stroke, and the glucose and oxygen supplied to the ischemic region decrease drastically. In the stroke core, irreversible cell death occurs within minutes due to complete disruption of blood perfusion around the core region (stroke penumbra), with restricted perfusion and impaired function, which may either recover or progress to infarction over time. In addition, the main cerebral functional carrier neurons are relatively lacking in glycolysis and antioxidant enzymes, which determine the high sensitivity and vulnerability of the brain to hypoxia [41,42]. Therefore, hypoxia and reoxygenation constitute an important aspect of the injury mechanism of ischemic stroke and also become the key simulation direction in in vitro models.

Previous studies [35,43,44] have demonstrated that in vitro OGD/R models can largely reflect cerebral ischemic events at the cellular level. Combined with previous studies, OGD generally responds well to the stroke core zone, but it is not believed that neuronal cell ischemia models are located in the penumbra zone. In our study, we used a tri-gas incubator (94% N_2_/5% CO_2_/1% O_2_) to cause hypoxia in cells and then reoxygenated them for 24 h. Our findings verified that OGD/R causes severe damage to HT22 cells; we examined their morphology and discovered that the cells had become transparent. In addition, their synapses were shortened or even broken, and their adherent capacity had been diminished.

After the successful development of the OGD/R model, the cell survival rate post-OGD/R at different time points was detected by MTT, and it was discovered that the cell survival rate of the HT22 cells after OGD/R treatment gradually decreased with the extension of OGD time and dropped to about 50% of the control group at OGD-6h/R. Combined with the previously reported findings, most of the OGD/R time selected led to a reduction in cell viability to about 50% in the control group [45]. Under our laboratory conditions, to moderate the damage to HT22 cells, we selected OGD-6h/R for follow-up experiments. Subsequently, it was demonstrated that H_2_ could improve neuronal survival rate and reduce LDH level, cell apoptosis rate, and levels of pro-inflammatory factors post-OGD/R. We also found that the protective properties of H_2_ could be linked to the regulation of the lincRNA-EPS/Sirt1 signaling pathway and the inhibition of autophagic activity. In addition, we confirmed the hypothesis that H_2_ could inhibit autophagy by upregulating the lincRNA-EPS/Sirt1 signaling pathway to alleviate OGD/R injury using autophagy inhibitors and activators, as well as transfecting siRNA to knockdown lincRNA-EPS. Consequently, the present study revealed that H_2_ plays a protective role against OGD/R damage in HT22 cells by upregulating the lincRNA-EPS/Sirt1 signaling pathway and inhibiting autophagy.

CI/R injury is associated with multiple brain disorders and can be fatal or disabling. CI/R injury can also cause a series of pathological reactions, including the production of oxygen radicals, calcium ion overload within the cell, glutamate-induced neurotoxicity, inflammatory response, cell apoptosis, and autophagy [2,5,6]. Ischemic preconditioning is a potent protective strategy against CI/R injury [46]. Interestingly, many studies have also shown that pharmacological preconditioning (PreCo) or postconditioning (PoCo) is beneficial to the ischemic organ. Subsequently, the main advantage of pharmacological PoCo over ischemic preconditioning or pharmacological PreCo is its added clinical feasibility or clinically applicable [47]. CI/R injury is often seen in unannounced clinical situations, such as cardiac arrest or ischemic stroke. Because CI/R is unforeseeable and inevitable in these situations, PoCo with hydrogen after CI/R appears to be feasible and effective.

Numerous studies have shown that hydrogen possesses antioxidant, anti-inflammatory, anti-allergic, and anti-apoptotic effects, and it can regulate autophagy levels [7,8]. Extensive research has demonstrated the protective function of hydrogen gas or hydrogen-rich saline on a variety of models of organ injury [9,10,11,12]. However, the signaling pathways and molecular mechanisms have remained elusive. Although some research reports indicate that hydrogen’s protective properties stem from its ability to stimulate autophagy, others have proven that its beneficial effects originate from its ability to suppress autophagy [48,49]. Our findings illustrated that after OGD/R injury, the cells were severely damaged, which was manifested by the reduced cell survival rate, increased apoptosis rate, elevated levels of LDH and inflammatory factors, and activation of autophagy. After treatment with hydrogen, the cell survival rate increased, and the LDH level, apoptosis rate, levels of pro-inflammatory factors, and autophagic rate were remarkably reduced, demonstrating that H_2_ could improve the neuronal OGD/R injury by suppressing autophagy.

Autophagy is an intracellular degradation system that delivers cytoplasmic constituents to the lysosome [28,30]. The Beclin-1 and LC3 were used to detect autophagy in the present study. Beclin-1 has sequence similarity with the yeast Atg6 gene in mammals, which may bind to class III phosphatidylinositol triphosphokinase (Class III PI3K), form a complex, contribute to the formation of autophagosomes, and regulate autophagy; a higher expression level of Beclin-1 indicates a stronger autophagic activity of the cells [50]. LC3 is the signature protein of autophagy, and when autophagy is initiated, cytoplasmic LC3 (LC3-I) changes from cytoplasmic recruitment to autophagosome membrane to LC3-II [51]. Therefore, the occurrence of autophagy can be confirmed by detecting the levels of Beclin-1 and LC3. Our findings illustrated that OGD/R damage can lead to autophagy activation, reduced cell survival rate, increased cell apoptosis rate, and elevated levels of LDH and pro-inflammatory factors. These injuries are significantly worsened when the autophagy is over-activated by autophagic agonists, and the injuries are relieved when autophagic inhibitors are used. Our study indicated that the neuronal protection of H_2_ was abolished by RAPA, in which cell apoptosis rate, LDH level, and levels of pro-inflammatory factors were remarkably elevated, while the cell survival rate was remarkably reduced. The treatment with 3-MA also confirmed the above-mentioned results. Taken together, it was shown that OGD/R activates autophagy and causes severe cell injury, while H_2_ can reduce cell injury caused by OGD/R by inhibiting autophagy.

The molecular mechanisms involved in autophagy regulation are very complex and include multiple signaling pathways [27,30]. Histone deacetylase Sirt1 requires NAD+ for its functioning and is implicated in the modulation of numerous biological activities, such as mitochondrial biogenesis, inflammatory response, oxidative stress, energy homeostasis, cell apoptosis, and autophagy in I/R injury. An earlier research report illustrated that hydrogen reduces hepatocyte apoptosis by stimulating the HO-1/Sirt1 pathway in the hepatic I/R injury model [26]. In addition, numerous studies have confirmed that autophagy was inhibited by stimulating the Sirt1 pathway, thus exerting a protective role [27,52]. Multiple investigations have shown that lincRNA-EPS modulates autophagy and has anti-inflammatory and anti-apoptotic properties [21,22,53]. A recent report showed that the exogenous administration of lincRNA-EPS inhibited microglial activation, reduced cytotoxicity and apoptosis, elevated the proliferative rate of neural stem cells in vitro, decreased the permeability of inflammatory cells, and improved neuronal cell death in ischemic regions in animal models of CI/R [54]. In our previous study, we discovered that neuronal lincRNA-EPS gene expression level was elevated following OGD/R, whereas Sirt1 mRNA and protein levels were reduced, and Beclin-1 and LC3II levels were increased; after H_2_ was introduced, however, lincRNA-EPS gene expression was further enhanced, Sirt1 mRNA and protein levels were upregulated, and Beclin-1 and LC3II mRNA and protein levels were downregulated. Therefore, we speculate that H_2_ has the potential to inhibit autophagy by activating the lincRNA-EPS/Sirt1 signaling pathway and playing a cytoprotective role in OGD/R. To confirm this inference, we used siRNA to knock down the lincRNA-EPS gene and found that after the lincRNA-EPS knockdown, the protective effect of H_2_ was partially weakened. Additionally, it was found that Sir1 was downregulated after the knockdown of lincRNA-EPS, and the autophagic activity of cells was also reduced, indicating that H_2_ could inhibit autophagy by activating the lincRNA-EPS/Sirt1 signaling pathway and alleviating cell injury resulting from OGD/R.

The current research has several drawbacks. Firstly, it is notable that in vivo tests were not performed. Interestingly, many previous studies have also used cell experiments alone to investigate brain injury [33,43,55]. Therefore, the experiments in vitro can reveal the related mechanism of the neuroprotection of hydrogen at a certain level. We will also expand the model using HT22 cells that go through differentiation conditions in our future research [56]. Secondly, overexpression of lincRNA-EPS was not performed, thus we did not fully confirm that H_2_ plays a role in activating the lincRNA-EPS/Sirt1 signaling pathway. Thirdly, we did not assess the regulatory relationship between Sirt1 and autophagy. Last but not least, this study was only a basic experiment, and its clinical effectiveness needs further verification.

## 5. Conclusions

In summary, our study highlighted the crucial function of the lincRNA-EPS/Sirt1/autophagy-dependent pathway in the attenuation impact of H_2_ on OGD/R-triggered injury in hippocampal neurons. Our findings further illustrated that lincRNA-EPS could be a possible target for H_2_, and we established a novel experimental foundation for the H_2_ therapy of CI/R injury (Figure 10).

## Figures and Tables

**Figure 1 jpm-13-00631-f001:**
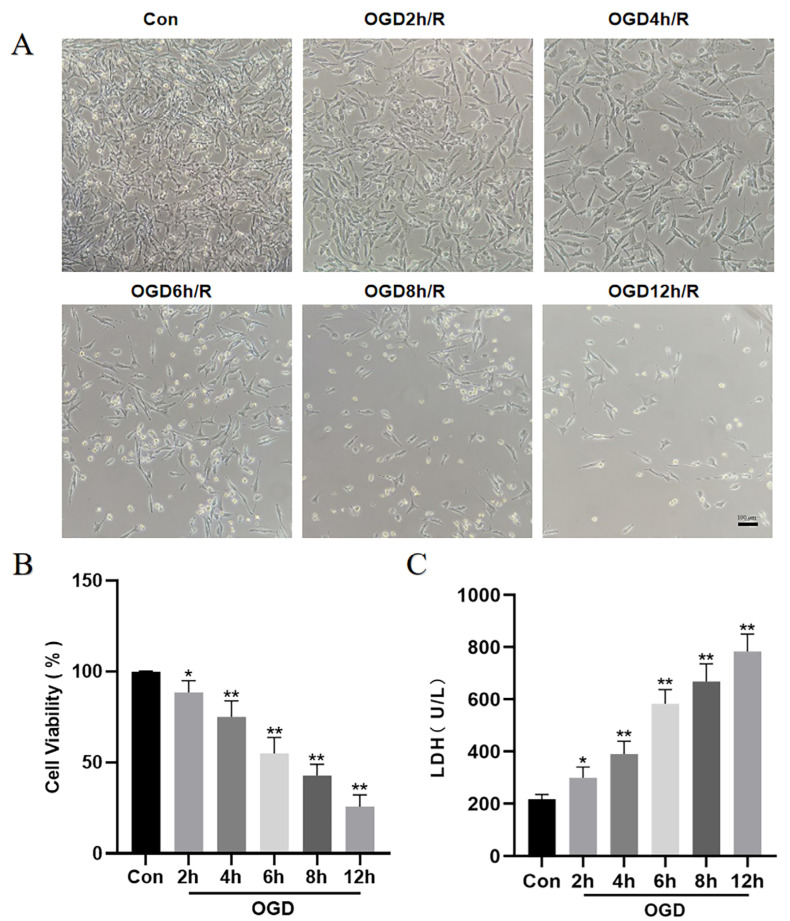
The different OGD time points on cell viability. (**A**) Live cell images captured by a light microscope from HT22 cells subjected to OGD treatments of varying intervals (2-, 4-, 6-, 8-, or 12-hrs), after which reoxygenation continues for another 24 h (×100; scale bar, 100 μm). (**B**) Viability of cells post-OGD at different time frames, accompanied by reoxygenation (MTT assay). (**C**) The release of LDH in each group. Statistics were reported using a mean ± SD format. A minimum of three separate repetitions of the experiments yielded the same results. * *p* < 0.05, ** *p* < 0.01 vs. Con group.

**Figure 2 jpm-13-00631-f002:**
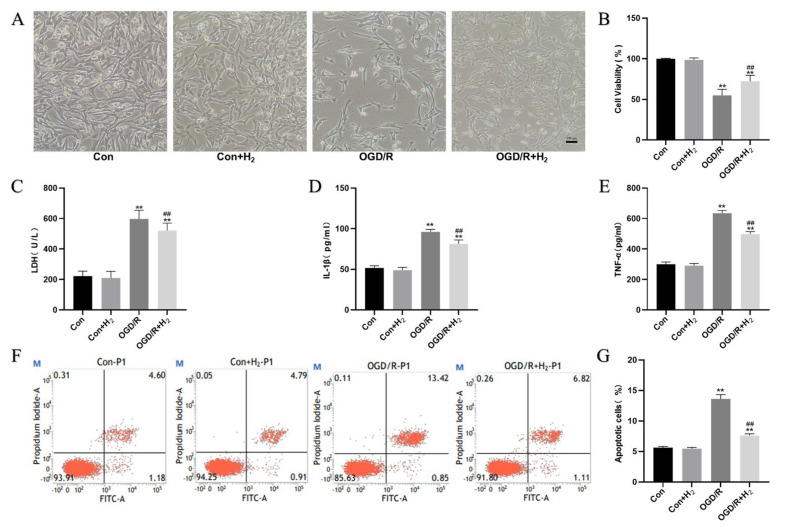
Neuroprotective effects of H_2_ against OGD/R injury. (**A**) Representative light microscopy of HT22 cells treated with or without hydrogen post-OGD/R injury. (**B**) The influence of H_2_ on neuronal viability of HT22 cells post-OGD/R injury. (**C**) The release of LDH in each group. (**D**) The IL1-β level in the medium. (**E**) The TNF-α level in the medium. (**F**) Images of HT22 cell analysis by flow cytometry as shown by dot plots. (**G**) Flow cytometry was used for the quantitative analysis of the apoptotic rate. The data were displayed as mean ± SD. Three independent runs of the experiments yielded similar results. ** *p* < 0.01 vs. Con group. ^##^
*p* < 0.01 vs. OGD/R group.

**Figure 3 jpm-13-00631-f003:**
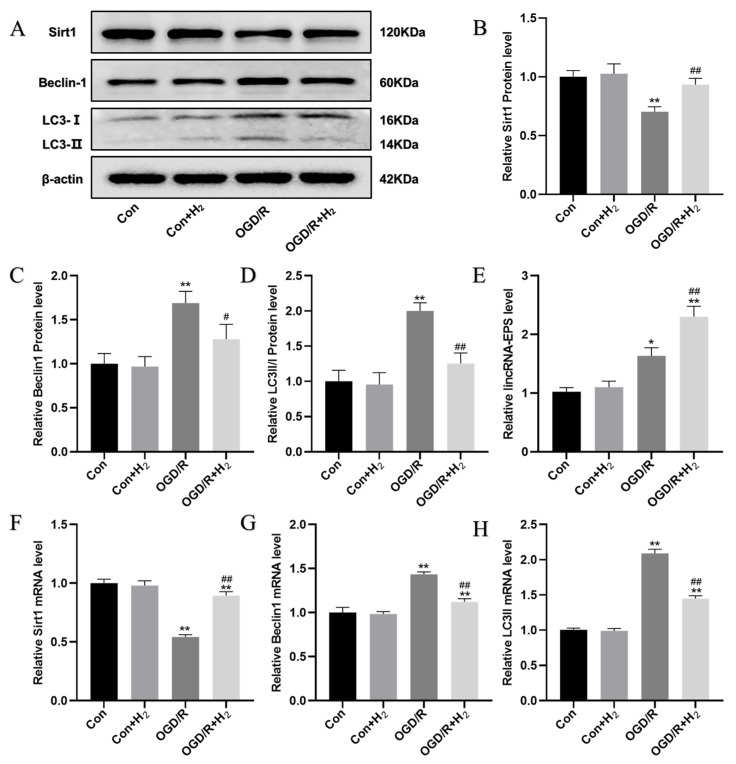
The possible protective mechanism of H_2_ against neuronal OGD/R injury. (**A**) Sample bands from a Western blotting experiment. (**B**–**D**) Quantification of Sirt1, Beclin-1, and LC3-II proteins in each group, respectively. (**E**) The levels of lincRNA-EPS expression were upregulated by H_2_ under OGD/R injury. (**F**) H_2_ upregulated the mRNA level of Sirt1. (**G**,**H**). The mRNA levels of LC3-II and Beclin-1 per group. Data were displayed as mean ± SD. A minimum of three separate repetitions of the experiments produced similar results. * *p* < 0.05, ** *p* < 0.01 vs. Con group. ^#^
*p* < 0.05, ^##^
*p* < 0.01 vs. OGD/R group.

**Figure 4 jpm-13-00631-f004:**
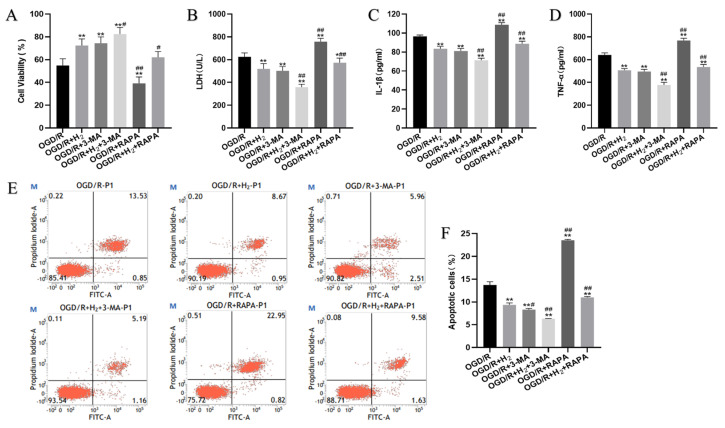
H_2_ protected HT22 cells from OGD/R injury by inhibiting autophagy. (**A**) The neuronal viability of HT22 cells per group. (**B**) The release of LDH in each group. (**C**) The IL1-β level in the medium. (**D**) The TNF-α level in the medium. (**E**) Images of HT22 cell analysis by flow cytometry as depicted by dot plots. (**F**) Quantification of the cellular apoptotic rates via flow cytometry. A minimum of three separate repetitions of the experiments produced similar findings. * *p* < 0.05, ** *p* < 0.01 vs. OGD/R group. ^#^
*p* < 0.05, ^##^
*p* < 0.01 vs. OGD/R+H_2_ group.

**Figure 5 jpm-13-00631-f005:**
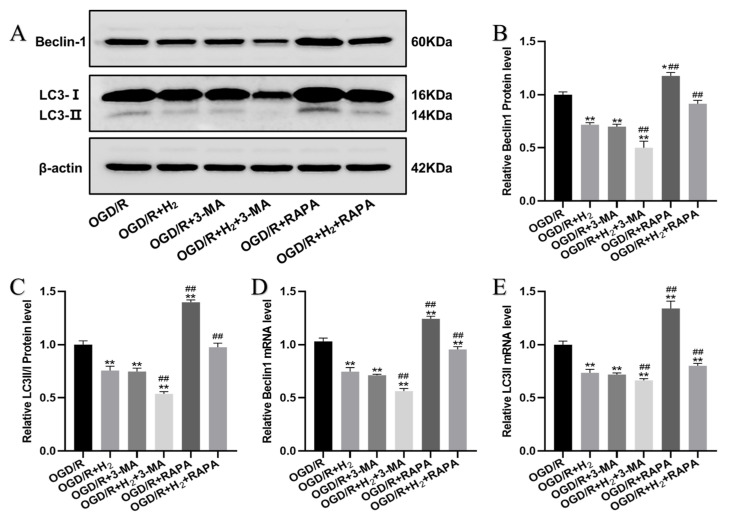
The effect of H_2_ on autophagy. (**A**) Sample bands from a Western blotting experiment. (**B**,**C**) Quantification of Beclin-1 and LC3-II proteins per group, respectively. (**D**,**E**). Levels of Beclin-1 and LC3-II mRNA expression per group. A minimum of three separate repetitions of the experiments produced similar findings. * *p* < 0.05, ** *p* < 0.01 vs. OGD/R group. ^##^
*p* < 0.01 vs. OGD/R+ H_2_ group.

**Figure 6 jpm-13-00631-f006:**
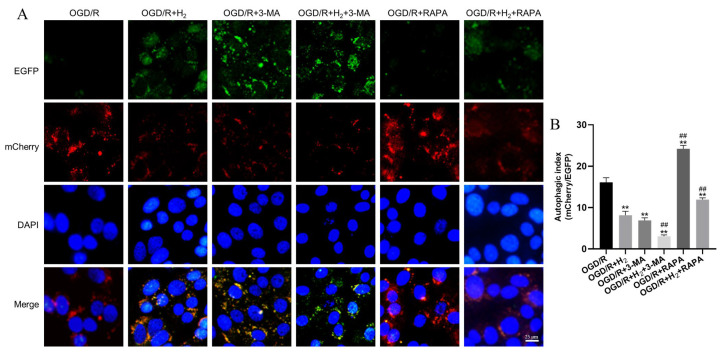
The changes of autophagic flux after OGD/R injury. (**A**) The mCherry-EGFP-LC3B reporter was transfected into HT22 cells for stable expression. (**B**) Quantifying the autophagic flow was achieved by comparing the mCherry+ puncta to the EGFP+ puncta (autophagic index). The mean autophagic index was plotted, with each data point corresponding to a field consisting of a minimum of 25 examined cells (Scale bar: 25 μm). Data were reported using a mean and standard deviation format. A minimum of three separate repetitions of the experiments produced similar findings. ** *p* < 0.01 vs. OGD/R group. ^##^
*p* < 0.01 vs. OGD/R+ H_2_ group.

**Figure 7 jpm-13-00631-f007:**
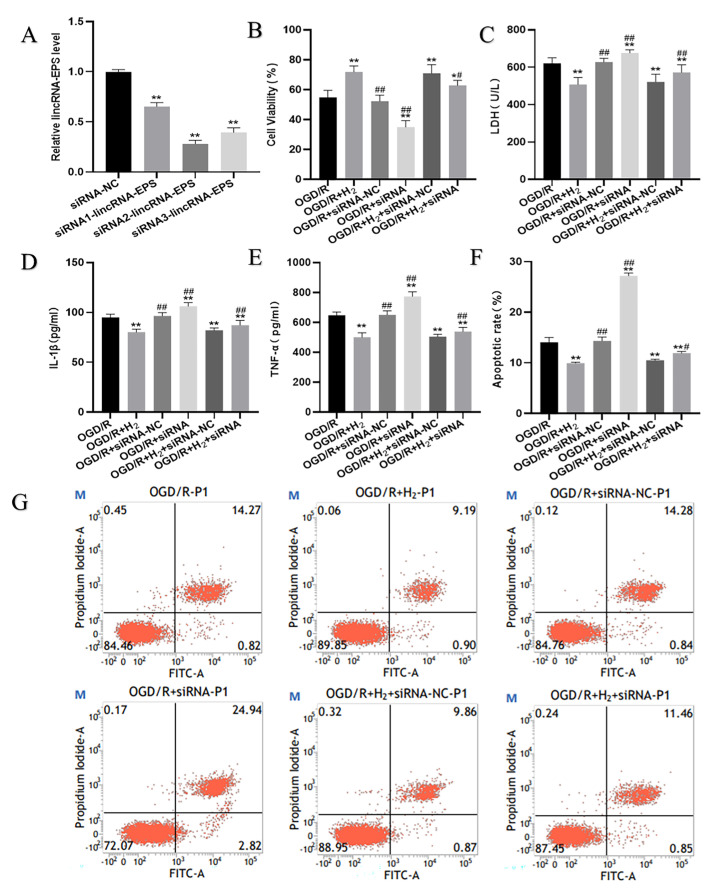
LincRNA-EPS knockdown weakened the protective effects of H_2_. (**A**) The siRNA-lincRNA-EPS efficacy evaluation. The data were displayed in a mean ± SD format. * *p* < 0.05, ** *p* < 0.01 vs. siRNA-NC group. (**B**) The cell viability in each group. (**C**) The release of LDH in each group. (**D**) The IL1-β level in the medium. (**E**) The TNF-α level in the medium. (**F**) Quantification of the apoptosis levels by flow cytometry. (**G**) Images of HT22 cell analysis by flow cytometry as represented by dot plots. Data were reported using a mean  ±  SD format. A minimum of three separate repetitions of the experiments produced similar findings. * *p* < 0.05, ** *p* < 0.01 vs. OGD/R group. ^#^
*p* < 0.05, ^##^
*p* < 0.01 vs. OGD/R+ H_2_ group.

**Figure 8 jpm-13-00631-f008:**
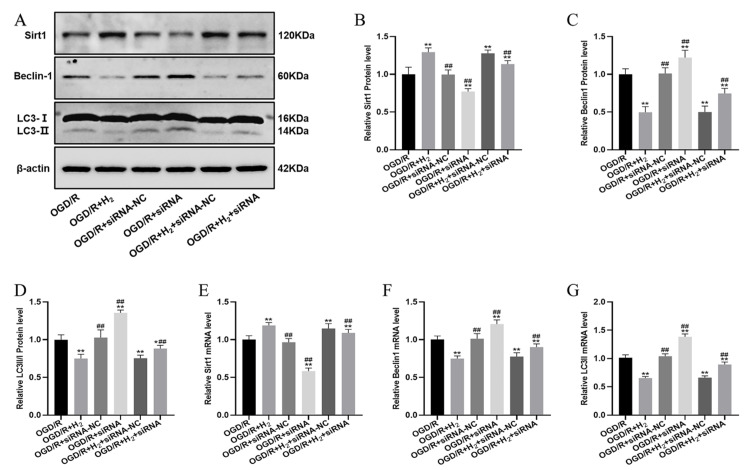
LincRNA-EPS knockdown abolished the effects of H_2_ on Sirt1 expression and autophagy. (**A**) Sample bands from a Western blotting experiment. (**B**–**D**) Quantification of Sirt1, Beclin-1, and LC3-II proteins in each group, respectively. (**E**–**G**). Sirt1, Beclin-1, and LC3-II mRNA expression levels were compared between groups. Data were reported using a mean  ±  SD format. A minimum of three separate repetitions of the experiments produced similar findings. * *p* < 0.05, ** *p* < 0.01 vs. OGD/R group. ^##^
*p* < 0.01 vs. OGD/R+ H_2_ group.

**Figure 9 jpm-13-00631-f009:**
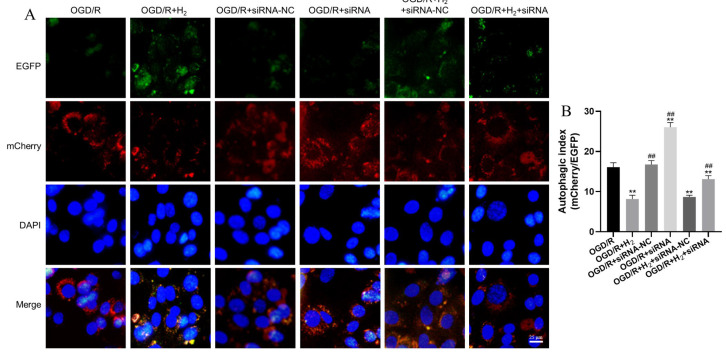
LincRNA-EPS knockdown abolished the effect of H_2_ on autophagic flux. (**A**) HT22 cells were transfected to stably express the mCherry-EGFP-LC3B reporter. (**B**) The ratio of mCherry^+^ puncta to EGFP^+^ puncta (autophagic index) was determined to quantify the autophagic flux. The mean autophagic index was plotted, with each data point corresponding to a minimum of 25 examined cells (Scale bar: 25 μm). Data were reported using a mean  ±  SD format. A minimum of three separate repetitions of the experiments produced similar findings. ** *p* < 0.01 vs. OGD/R group. ^##^
*p* < 0.01 vs. OGD/R+ H_2_ group.

**Figure 10 jpm-13-00631-f010:**
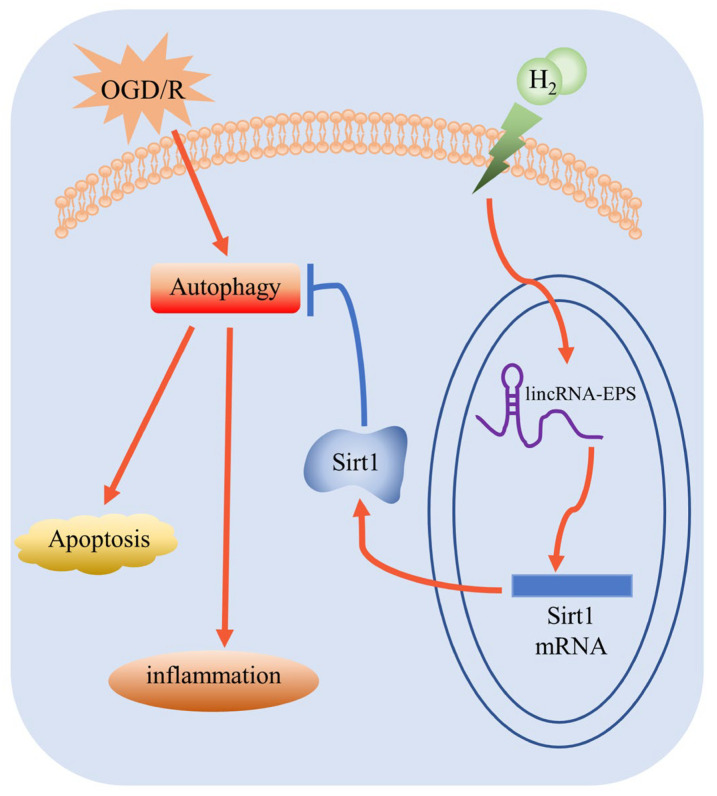
The molecular mechanism of H_2_ against OGD/R-induced injury.

**Table 1 jpm-13-00631-t001:** Summary of primer sequences used for RT-PCR.

Gene	Primer Sequences
lincRNA-EPS	Forward: CTCGATCTCACTGCATGGCT
	Reverse: TAGGATGGGAGGTAGTGCCA
Sirt1	Forward: TCACACGCCAGCTCTAGTGA
	Reverse: CAGCTCAGGTGGAGGAATTGT
Beclin1	Forward: AGGCATGGAGGGGTCTAAGG
	Reverse: AATGGCTCCTGTGAGTTCCTG
LC3-II	Forward: TTATAGAGCGATACAAGGGGGAG
	Reverse: CGCCGTCTGATTATCTTGATGAG
GAPDH	Forward: TGGCCTTCCGTGTTCCTAC
	Reverse: GAGTTGCTGTTGAAGTCGCA

## Data Availability

Data are contained within the article.

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
