# Peer review of "The Role of LincRNA-EPS/Sirt1/Autophagy Pathway in the Neuroprotection Process by Hydrogen against OGD/R-Induced Hippocampal HT22 Cells Injury"

_jpm, 2023, doi:10.3390/jpm13040631_

Round 1

Reviewer 1 Report

The article under review addresses a clinically relevant question of the molecular mechanisms underlying the previously reported beneficial effects of H2 in brain ischemia/reperfusion, though in my opinion it is a question, whether the presented paper falls within the scope of the journal and the special issue. Experimental design is mostly all right, except that the authors claim that it is difficult to read, and so the other sections of the article. The discussion lacks a part which in my opinion is necessary. I mean the relevance of the used in vitro model to neural cell injury in the real world ischemic brain. Is a model for neurons located in the penumbra zone?

Author Response

Thanks for very good comments and suggestion. Although in vitro experiments cannot fully reflect cerebral ischemic events, many previous studies have demonstrated that in vitro OGD/R models can largely reflect cerebral ischemic events at the cellular level [1-3]. Combined with previous studies [4-5], OGD generally responds well to the stroke core zone, but it is not believed that neuronal cell ischemia models are located in the penumbra zone, which needs to be further explored in the future.

We have also strengthened the discussion about our findings and have addressed this point as the limitation of the present study in the section of discussion. (Please see Discussion: Page 22, lines 658~685).

Reviewer 2 Report

Dear Authors,
After reading your paper, I have mentioned to the Editors that in my opinion your manuscript could be consider for publication after several very minor issues -enlisted below- are addressed:

 1.       In the third line of the Abstract, a space is missing between unclear and Long

2.       It is not necessary to state two times in the abstract that rapamycin is an autophagy agonist (lines 26 and 30); one time is OK. Same for line 146 vs line 154, which is also repeated in lines 330 and 491.

3.       In page 29 in the Abstract, perhaps the authors mean “insult” instead of “insulted”.

4.       The abstract mentions that Autophagy, Neuroinflammation and Apoptosis were evaluated. However, perhaps the term neuroinflammation should be treated a bit differently in the context of this work, as the cells used were HT22, which, at least differentiated, do not necessarily represent all the classical properties of functionally developed adult neurons. Therefore, and while the concept is clear and the intention of the authors is very appropriate, technically, neuroinflammation per se is not really evaluated, but rather the changes in the levels of molecules implicated in neuroinflammation (pro-inflammatory factors). Please briefly address this in the paper (in methods or discussion, etc.,) and/or change accordingly.

5.       In line 102: please check Font style and size.

6.       In line 118: while the concept is again clear, technically, “reperfusion” per se is not really taking place but rather “conditions resembling some important features of vascular reperfusion” or “reperfusion-like model”, etc. Please clarify this, so that confusions with the use of the concept reperfusion, as used, e.g., in "ischemia-reperfusion injury" (with more than 27.000 entries in PubMed) and the use of reperfusion to refer to a model of reperfusion are not confused.

7.       In line 167: Add a space in between “groups.After”. Same in line 290 for “group.Data”

8.       In line 167: Again, I suggest to use something like “reperfusion-like conditions” instead of “reperfusion”.

9.       Please include in the Methods published references, if available, for the use of siRNAs as a method to specifically inhibit lincRNA-EPS.

10.   Page 7. Please change the format of figure legend of Table 1 for better legibility.

11.   In line 394: Add a space in between “evaluation.Data”.

12.   In line 456, add space before “Ischemic preconditioning”

13.   Line 525: add a spaces at “Therefore,the”

That is all from my side: I look forward to reading your final manuscript published in the prestigious JPM.

Note. With independence of the decision of the Main Editors, I also believe that - in your future work-, it would be also very interesting to expand your model using also HT22 cells that go through differentiation conditions (see for example He, et.al., Neural Regen Res. 2013 May 15; 8(14): 1297–1306.).

Author Response

Response: Thanks so much for the reviewer’s good suggestions. We totally agree with the reviewer's opinion and we have revised it one by one. We have used “reoxygenation” instead of “reperfusion” according to some previous studies [6-8]. In addition, a published reference has been cited for LincRNA-EPS siRNA [9]. The above details were addressed in the revised manuscript.

    Based on the reviewer’s good suggestions, we have read this article in detail (He, et.al., Neural Regen Res. 2013 May 15; 8(14): 1297–1306.) and have strengthened the discussion combined with it. We added these comments in discussion section in the revised manuscript. (Please see Discussion: Page 26, lines 807~808).
